# Region-Specific Decellularization of Porcine Uterine Tube Extracellular Matrix: A New Approach for Reproductive Tissue-Engineering Applications

**DOI:** 10.3390/biomimetics9070382

**Published:** 2024-06-24

**Authors:** Gustavo Henrique Doná Rodrigues Almeida, Raquel Souza da Silva, Mariana Sversut Gibin, Victória Hellen de Souza Gonzaga, Henrique dos Santos, Rebeca Piatniczka Igleisa, Leticia Alves Fernandes, Iorrane Couto Fernandes, Thais Naomi Gonçalves Nesiyama, Francielle Sato, Mauro Luciano Baesso, Luzmarina Hernandes, Jaqueline de Carvalho Rinaldi, Flávio Vieira Meirelles, Claudete S. Astolfi-Ferreira, Antonio José Piantino Ferreira, Ana Claudia Oliveira Carreira

**Affiliations:** 1Department of Surgery, School of Veterinary Medicine and Animal Science, University of São Paulo, São Paulo 03828-000, Brazil; raquellsouzasilva@outlook.com (R.S.d.S.); leticia.afernandes@usp.br (L.A.F.); ranecouto@gmail.com (I.C.F.); ancoc@iq.usp.br (A.C.O.C.); 2Department of Physics, State University of Maringá, Maringá 87020-900, Brazil; marigibin32@gmail.com (M.S.G.); victoriahsgonzaga@gmail.com (V.H.d.S.G.); rique.lovo@gmail.com (H.d.S.); fsato@uem.br (F.S.); mlbaesso@dfi.uem.br (M.L.B.); 3The Ken & Ruth Davee Department of Neurology, Northwestern University Feinberg School of Medicine, Chicago, IL 60611, USA; rebeca.iglesia@northwestern.edu; 4Department of Veterinary Medicine, Faculty of Animal Science and Food Engineering, University of São Paulo, São Paulo 05508-270, Brazil; thais_ngn@usp.br (T.N.G.N.); meirellf@usp.br (F.V.M.); 5Department of Morphological Sciences, State University of Maringá, Maringá 87020-900, Brazil; lhernandes@uem.br (L.H.); jak.rinaldi@gmail.com (J.d.C.R.); 6Department of Pathology, School of Veterinary Medicine and Animal Science, University of São Paulo, São Paulo 05508-270, Brazil; csastolfi@gmail.com (C.S.A.-F.); ajpferr@usp.br (A.J.P.F.); 7Centre for Natural and Human Sciences, Federal University of ABC, Santo André 09040-902, Brazil

**Keywords:** uterine tube, extracellular matrix, reproduction, decellularization, biomaterial

## Abstract

The uterine tube extracellular matrix is a key component that regulates tubal tissue physiology, and it has a region-specific structural distribution, which is directly associated to its functions. Considering this, the application of biological matrices in culture systems is an interesting strategy to develop biomimetic tubal microenvironments and enhance their complexity. However, there are no established protocols to produce tubal biological matrices that consider the organ morphophysiology for such applications. Therefore, this study aimed to establish region-specific protocols to obtain decellularized scaffolds derived from porcine infundibulum, ampulla, and isthmus to provide suitable sources of biomaterials for tissue-engineering approaches. Porcine uterine tubes were decellularized in solutions of 0.1% SDS and 0.5% Triton X-100. The decellularization efficiency was evaluated by DAPI staining and DNA quantification. We analyzed the ECM composition and structure by optical and scanning electronic microscopy, FTIR, and Raman spectroscopy. DNA and DAPI assays validated the decellularization, presenting a significative reduction in cellular content. Structural and spectroscopy analyses revealed that the produced scaffolds remained well structured and with the ECM composition preserved. YS and HEK293 cells were used to attest cytocompatibility, allowing high cell viability rates and successful interaction with the scaffolds. These results suggest that such matrices are applicable for future biotechnological approaches in the reproductive field.

## 1. Introduction

The female reproductive tract microenvironment plays a direct and essential influence on embryonic morphogenesis, from the oocyte release in the ovarian follicle to the conceptus implantation in the endometrium, modulating cellular processes for the formation of a new individual [1]. Although the endometrium is considered the main tissue for the proper establishment of maternal–embryo interaction, promoting communication between the maternal blood system and the placenta, the uterine tube is actually the first microenvironment that the newly fertilized embryo interacts with [2]. During this process, the tubal epithelium inhibits polyspermy, and assists on spermatic capacitation and flagellar movement, through ciliary motility and the secretion of a fluid enriched with nutrients and signaling molecules [3]. On the other hand, the tubal microenvironment specifically in the ampulla region acts on the genome activation of embryonic cells and influences the quality of inner mass cells, and such interaction contributes to embryonic survival [4].

Despite the importance of uterine tubes in natural conception in mammals, this structure became non-essential for the in vitro fertilization (IVF) techniques process, once the embryos can be produced in vitro and transferred to the maternal endometrium after a short period in culture, even for human samples [5,6]. Nevertheless, even with the advances in in vitro assisted reproduction techniques, live birth rates after embryo transfer are lower than expected, leading to a high percentage of gestational loss and an increase in the expenses of IVF [7]. According to the Society for Reproductive Technology (SART), the successful birth rate in women under 35 who have undergone IVF in the United States is only 55.6% [8] and the percentage decreases at more advanced ages, such as with 38–40-year-old women [8]. In other species on which IVF is practiced because of economic interest, such as bovines, the success rate of embryos transferred after IVF is only 38%, which is much lower compared to the natural mating process [9].

In this context, studies have focused on elucidating the molecular mechanisms presented in the female reproductive tract and their influence on embryo development and in the implantational process [10,11]. The tubal microenvironment has gained more protagonism in pregnancy success, once the interaction between the embryo and the tubal epithelium established the basis for embryonic development [12]. Indeed, the molecular communication that ciliated epithelial cells play on germ cells and in the newly formed embryo is able to modulate phenotypic characteristics and induce important modifications in cellular processes, contributing to fertilization and, consequently, implantation success [13].

The extracellular matrix (ECM) of the tubes provides not only structural support to the cells, but also is able to self-modulate due to cyclical hormonal stimulation in the female reproductive tract [14,15]. The tubal ECM molecular signaling is able to stimulate cell proliferation in the mucosa through mechanotransduction, as well as their differentiation in the several specialized cell types along the tubal epithelium and stroma [16]. Mechanic forces that the tubal wall plays on the embryo, such as shear stress and compression forces, also modulate cell fate and contribute to embryonic development and survival [17,18]. This complex microenvironment demonstrates that a better understanding of its cellular and molecular constituents may provide important data for the development of new approaches for early infertility diagnosis, as well as the improvement of in vitro reproduction techniques to increase successful pregnancy rates [19,20].

The study of early development events is very difficult in humans, once the management of human samples in the pre-implantational phase is limited for sample collection due to a very short and dynamic phase, and also, there are several technical and ethical parameters involved in the studies of human embryos [21]. Thus, the production of in vitro models that recapitulate the tubal morphophysiology to study the interaction between the embryo and the components that structure the tubal wall under controlled parameters [21,22] has been proposed. A number of artificial 2D and 3D methods were developed to mimic the ciliated secretory epithelium, such as the transwell systems and tubal organoid; however, none of them were able to recapitulate the uterine tube histoarchitecture [21,23]. Cell–cell and cell–ECM contact is essential to structure a functionally appropriated biomimetic model that allows co-culture with embryos or embryonic organoids [23].

Although these models aim to rebuild portions of the tubal wall, cell–cell and cell–environment interactions are essential to mimic in vivo conditions [24,25]. To develop more complex models, the use of natural and synthetic biomaterials that may provide a functional three-dimensional structure that allows cellular adhesion and maintenance to reconstitute the tissue microarchitecture [14,15] has been proposed. Several natural biomaterials such as collagen, alginate, Matrigel^®^, and synthetic ones such as polyethylene glycol (PEG) and polycaprolactone have a number of mechanical, physical–chemical, and biological properties for application in biomimetic tissue reconstruction; however, none of them are able to mimic efficiently and precisely the ECM properties [26]. Tubal ECM bioactive peptides present specific characteristics and properties that are singular to the uterine tube microenvironment [16]; thus, obtaining biological matrices by decellularization is an innovative and accessible approach that allows the preservation of ECM composition and three-dimensionality [27,28]. Recent studies demonstrated that biological matrices are able to support cells from several biological sources and species, revealing their xenogeneic compatibility [29,30]. The application of biological scaffolds derived from decellularized tubal matrices has great biotechnological potential, not only for tubal microenvironment reconstruction, but also as a substrate to stimulate oocyte maturation and in vitro embryo development from different species [27,28].

Therefore, this study aimed to establish region-specific decellularization protocols to produce biological acellular extracellular matrices from porcine infundibulum, ampulla, and isthmus to access the preservation of the main ECM components for the development of high-quality porcine tubal matrices. Structural and physico–chemical analyses were performed to evaluate the efficiency of each presented protocol, and the cytocompatibility of the scaffolds was also verified.

## 2. Materials and Methods

### 2.1. Sample Acquisition

Uterine tubes (*n* = 10 per group) from pre-pubertal sows at approximately 6 months of age were obtained from the slaughterhouse of the Faculty of Animal Science and Food Engineering of the University of São Paulo, Pirassununga campus, Brazil. Immediately after collection, the samples were stored on ice and transported to the laboratory. The methodology is summarized in Figure 1.

### 2.2. Uterine Tube Segment Decellularization

The uterine tubes were dissected and divided into infundibulum, ampulla, and isthmus. Each fragment had a size of 1 cm, and they were submitted to the decellularization process. The decellularization process took place in three steps, varying according to the segment. For infundibulum and ampulla the steps included the following: (1) initial washing—2 h of dH_2_O and 2 h of sodium phosphate buffer (PBS) (137 mM NaCl, 10 mM phosphate, 2.7 mM KCl; pH 7.4) at 100 rpm of agitation; (2) decellularization—sodium dodecyl sulphate (SDS) 0.1% for 10 h and Triton X-100 0.5% for 2 h. For the isthmus, there were no differences in step 1, and step 2 involved longer exposure to SDS 0.1%, for 24 h. Samples were stored in PBS 1x at 4 °C for further analysis.

### 2.3. 4,6-Diamidino-2-Fenilindole (DAPI) Staining

DAPI staining was used to verify the presence of nuclei in the samples after the decellularization. Native and decellularized samples were frozen in Tissue Plus O.C.T. (Fisher Health Care, Houston, TX, USA), then microsectioned using a cryostat (CM1860 model, Leica Biosystems, Baden-Wurttemberg, Germany). The slides were stained with DAPI solution (1:10,000) at room temperature without light for 10 min, and then washed with PBS 1x for analysis using fluorescent microscopy (Nikon ECLIPSE 80I, CADI FMVZ-USP, Tokyo, Japan).

### 2.4. Genomic DNA Quantification

For genomic DNA quantification, the BioGene^®^ K204-4 (BioGene, Recife, Brazil) was used to extract the DNA from native and decellularized samples according to the manufacturer’s specifications. The fragments were digested at 56 °C by the action of Proteinase K. The fragments were filtered and analyzed by spectrophotometry at 260 nm (Nanodrop, Thermo Scientific, Waltham, MA, USA).

### 2.5. Histological Analysis

Histological evaluation was conducted to assess the decellularization efficiency and to verify the morphological integrity of ECM components. Native and decellularized tubal fragments were fixed in 4% buffered paraformaldehyde (PFA) for 48 h, dehydrated using sequential concentrations of alcohol (70, 80, 90, and 100%), diaphanized in xylol, and embedded in paraffin. Microsections of 5 µm (n° RM2265; Leica) were stained with hematoxylin and eosin (HE) to verify nuclei presence and the general tissue morphology; Masson’s trichrome for collagen content evaluation; Picrosirius red to evaluate the pattern of collagen distribution of different thicknesses; Alcian blue (pH = 2.5) to evaluate general content of GAGs and Weigert’s fuchsin–resorcin to detect elastic fiber presence. Slides were photographed and analyzed using a light microscope (Nikon ECLIPSE 80I, CADI FMVZ-USP). To perform measurements, the relative occupied area of these ECM elements was quantified in five semiseries slices per animal, totaling 10 fields at a magnification of 20 times with the software ImageJ version 8.0.

### 2.6. Immunohistochemistry Analysis

At first, sample microsections were rehydrated in citrate buffer in the microwave for antigen retrieval for 1 min. Then, the endogenous peroxidase blockage was performed with 3% hydrogen peroxide in distilled water for 40 min in the dark. After that, for the non-specific protein interaction blockage, 2% bovine serum albumin (BSA) in PBS was used. The primary antibodies chosen were anti-collagen I (#PA5-29569, 1:250, Invitrogen, Carlsbad, CA, USA), anti-collagen III (#PA1-28870, 1:250; Invitrogen), anti-fibronectin (#Ab2413, 1:100, Abcam, Cambridge, UK), anti-laminin subunit α2 (#PA1-16730, 1:200; Invitrogen), anti-elastin (#Ab9519, 1:100, Abcam), and the secondary antibodies were IgG anti-mouse/anti-rabbit (#K800; Dako, CA, USA). The incubation took place overnight at 4 °C. The reaction was detected by Dako Advance HRP (#K6068; Dako) followed by DAB (#k3468; Dako) revelation, according to the manufacturer’s instructions. Slides were photographed and analyzed using a light microscope (Nikon ECLIPSE 80I, CADI FMVZ-USP). To perform a measurement, the relative occupied area of these main ECM components was quantified in five semiseries slices per animal, totaling 10 fields at a magnification of 20 times with the software ImageJ version 8.0.

### 2.7. Scanning Electronic Microscopy (SEM)

Samples were fixed in Karnovsky solution (2.5% glutaraldehyde and 4% paraformaldehyde in a buffered 0.1 M sodium cacodylate) for 48 h and dehydrated in alcohol concentrations for 5 min each. Then, the fragments were dried in a supercritical point device (LEICA EM CPD 300^®^, Automated Critical Point Dryer, Leica Microsystems, Buffalo Grove/IL, USA) and coated with gold (EMITECH K550^®^, Quorum Technologies, Laughton, East Sussex, UK). Lastly, the samples were photographed under a scanning electron microscope (LEO 435 VP^®^, Oberkochen, Germany).

### 2.8. Fourier Transform Infrared Spectroscopy (FTIR) Analysis

To evaluate the ECM molecular composition of native and decellularized samples and verify if the main proteins remained preserved, Fourier transform infrared spectroscopy (FTIR) was applied. The assay was conducted in a Bruker Vertex 70v FTIR spectrometer (Bruker Optik GmbH, Ettilingen, Germany) with an attenuated total reflectance (ATR) accessory located at the Department of Physics from the State University of Maringá, Paraná, Brazil. The spectrum of each uterine tube segment (*n* = 5), between 4000 and 400 cm^−1^, is an average of three measurements with 128 scans and 4 cm^−1^ of spectral resolution. All measurements were performed at room temperature, and the spectra were vector normalized using OPUS software 8.5.

### 2.9. Raman Spectroscopy Analysis

To complement the FTIR analysis, Raman spectroscopy was also applied to evaluate physical–chemical alterations in native and decellularized tissues. Raman spectra were obtained using a Senterra Confocal Raman microscope (Bruker Optik GmbH, Ettilingen, Germany) equipped with an objective lens (20× magnification) focusing the excitation laser (785 nm, 100 mW) on the sample. The spectrum of each tubal segment, from 1750 to 400 cm^−1^, is an average of three measurements with three scans, 9–15 cm^−1^ of spectral resolution, and 20 s of detector integration time. All measurements were performed at room temperature and the spectra were vector normalized using OPUS software 8.5.

### 2.10. Scaffold Sterilization

At the end of decellularization, the scaffolds were washed with dH_2_O for 48 h to remove the detergents. For sterilization, inside the laminar flow hood, the scaffolds were fragmented and immersed in progressive and regressive alcohol concentrations (70%, 80%, 90%, 100%, 90%, 80%, and 70%) for 5 min in each one. After that, the segments were washed with PBS 1x with 2% antibiotics (Penicillin-Streptomycin 10,000 μg/mL, LGC Biotecnologia, Cotia, Brazil). Then, they were also exposed to UV light for 5 min and stored for further analysis. To evaluate the scaffolds’ sterility, they were immersed in DMEM medium (Sigma, St. Louis, MO, USA) without antibiotics and incubated for 72 h in the cell culture incubator.

### 2.11. Cytotoxicity Assay

To evaluate cell viability in the biomaterials, a resazurin assay was conducted. At first, canine Yolk sac-derived (YS) cells, which were isolated, characterized, and donated by Professor Ana Claudia Oliveira Carreira from University of São Paulo [31], and HEK293 cells were cultured in DMEM medium supplemented with 10% fetal bovine serum (FBS, Gibco, UK) and antibiotics, and incubated at 37 °C under 5% CO_2_ conditions. After 18 h, the cells were tripsinized and a density of 2 × 10^3^ cells/mL was seeded on tubal scaffolds in a 24-well plate. Then, 1 mL of resazurin solution (0.14 mg/mL in PBS; Thermo Fisher, Waltham, MA, USA) was added to each well. The seeded scaffolds were incubated for 1, 3, 7, and 10 days. On each day, 200 μL were collected and stored in a 96-well plate in the refrigerator. At the end of the experiment, the collected samples were read in a spectrophotometer (µQuant—Bio-Tek Instruments, INC) at an optical density of 570 nm. The same culture conditions were applied for cells cultured without the scaffolds, which were the control groups.

### 2.12. Cytocompatibility Assay

To verify the cell adhesion on the scaffolds, 1 × 10^4^ YS and HEK293 cells were seeded on infundibulum, ampulla, and isthmus fragments immersed in supplemented D-MEM medium for 7 days at the same culture conditions. Then, the seeded scaffolds were fixed for SEM analysis.

### 2.13. Statistical Analysis

For quantitative data, Shapiro–Wilk normality test was conducted to confirm data normal distribution. For DNA content data, Mann–Whitney test was applied. For morphoquantitative analyses, multiple Student’s t-tests were applied. Data were shown as mean ± SD (standard deviation). For FTIR and Raman experiments, a PCA analysis was conducted, and a one-way ANOVA was performed to compare the means of quantitative data. Statistical significance was considered for *p* < 0.05. A Tukey post-hoc test was applied to compare native and decellularized samples. The data were analyzed with GraphPad Prism 8.0 (GraphPad Software, Inc., San Diego, CA, USA).

## 3. Results

### 3.1. Tubal Segment Decellularization: Protocol Standardization

Considering that each tubal segment has a distinct structural composition, varying in thickness, more than one decellularization protocol was applied to allow the efficient cellular removal and maintenance of the structural scaffold, as well as the ECM composition of each segment. Several analyses were performed for each segment separately to verify their properties as a biological scaffold. Macroscopically, it was possible to observe a whitening in the infundibulum, ampulla, and isthmus compared to the native tissue with little alteration in the macroscopic structure (Figure 2A,C,D). At first, a unique protocol to decellularize the uterine tube as a whole was proposed; however, preliminary histological analyses showed that the exposure period to decellularize the infundibulum and the ampulla was insufficient to remove isthmus cellular content, which led to applying a longer exposure in the decellularizing solution. By DAPI staining analysis, it was verified that when compared to the native tissue, the 10 h SDS–based protocol for the infundibulum (Figure 2E,F) and for the ampulla (Figure 2G,H) was able to promote cellular removal efficiently. Regarding the isthmus, the same analysis showed that the 24 h SDS–based protocol was more efficient than decellularization (Figure 2I,J), since it presents a higher muscle layer thickness. Corroborating these findings, the total genomic DNA analysis showed a reduction in the DNA content of 97.6% in the infundibulum, 97.8% in the ampulla, and 94.7% in the isthmus compared to the native tissue, which demonstrated the decellularization efficiency (Figure 2K).

### 3.2. Morphological and Ultrastructural Characterization

The ECM components’ preservation after the decellularization process is vital in the development of biological scaffolds, once these elements structure the tissue microenvironment [32]. For this purpose, histological (Figure 3), immunohistochemical (Figure 4), and ultrastructural (Figure 5) analyses of the three decellularized tubal segments were conducted to perform a morphological and three-dimensional characterization of the generated biomaterials. Through hematoxylin and eosin (H&E) staining, it was possible to observe the tissues’ general structure, as well as the presence or absence of nuclei. In the native infundibulum (INF), there were evident epithelial invaginations toward the lumen, as well as several blood vessels, glandular structures, and a thin muscle layer (Figure 3A). In the decellularized infundibulum, it is possible to visualize an absence of nuclei, the maintenance of some invaginations, and rounded structures, corresponding to blood vessels and glands (Figure 3A^1^). Regarding the ampulla, there is a reduced number of glands, a more prominent lumen with fewer invaginations and a more intense presence of connective tissue (Figure 3A^2^). The decellularized ampulla (dAMP) also presented an absence of nuclei and maintained the vascular and glandular structures preserved, and it is possible to distinguish the regions with prominent connective tissue (Figure 3A^3^). In the isthmus (IST), the uterine tube muscular portion, there is a more reduced quantity of glands with a more stained connective portion, demonstrating a denser tissue (Figure 3A^4^). In the decellularized isthmus (dIST), the nuclei absence is also noteworthy, along with a more structured and dense extracellular matrix (Figure 3A^5^).

Total collagen content was evaluated using Masson’s trichrome staining, which highlighted the collagen fibers in dark blue (Figure 3B–B^5^). Morphologically, decellularized samples did not show any extracellular matrix degradation or abrupt disarrangement, being more evident in the dIST region, due to its characteristic of being highly vascularized (Figure 3B^5^). In dINF and dAMP, the morphology of the ECM presents minor alterations, which was expected due to the higher presence of loose connective tissue in these samples. The quantitative analysis of collagen fibers after the decellularization did not reveal any statistically significant difference between the native tissue and the decellularized scaffolds, demonstrating that there was no collagen degradation (Figure 3G). The means of the relative occupied area are summarized in Appendix A.

Additionally, Picrosirius red staining was used to highlight the collagen content distribution (Figure 3C–C^5^); under polarized light, thick collagen fibers present a red and yellow color while thin collagen fibers are green (Figure 3D–D^5^). Morphologically, there is a higher proportion of thin collagen fibers compared to the thick ones in the INF; in the AMP, this proportion is more balanced, although the thin fibers are still more abundant, while in the IST it is opposite, since the percentage of thick fibers is higher than thin fibers. This morphological pattern remains similar in the decellularized samples of the three segments; indeed, quantitative data corroborated the morphological findings, demonstrating no statistical difference for both thick and thin collagen fibers distribution between native and decellularized samples (Figure 3H,I).

The main non-fibrillary molecular group presented in the ECM is general glycosaminoglycan (GAG) content, which we evaluated using Alcian blue staining. Alcian blue highlights the GAG content in light blue in a diffuse pattern through the tissue (Figure 3E–E^5^); morphologically, GAGs had a similar distribution pattern in the three tubal segments both in native and decellularized samples. It is noteworthy that the decellularization process caused a mild degradation in the GAG content, since those molecules are more sensitive to the effects of detergents. However, the quantitative analysis did not evidence any statistical difference between the groups, which demonstrated that even with a small loss, the majority of the GAG content remained in the scaffolds after the decellularization process (Figure 3J).

The presence of elastic fibers was detected by Weigert’s fuchsin–resorcin staining, which stains the elastic components in intense dark purple (Figure 3F–F^5^). In the tubal samples, elastic fibers were more concentrated in the blood vessels, mainly in the elastic arteries and veins, with some fibers diffused through the non-modeled connective tissue. In the absence of cells, the elastic fibers became more evident, mainly in dAMP (Figure 3F^3^) and dIST (Figure 3F^5^). Quantitative analysis of the relative occupied area demonstrated an increase in elastic fibers in the dAMP and dIST samples compared to the native tissues (Figure 3K). This increase may be related to the absence of cells in the samples, which allowed the fibers to be more evident in the staining; however, there was no relevant statistical difference between the INF and dINF samples.

After a histological evaluation of tubal ECM, immunohistochemical analyses were performed to detect and quantify the main matrix proteins (Figure 4A–E^5^). Type I and III collagens, which are the main tubal structural components, presented a similar expression pattern between native and decellularized samples, with similar values in quantification (Figure 4F,G). Elastin, the main elastic fiber component, presented an increase in dAMP and dIST compared to native tissues, but no statistical difference was observed between the INF and dINF samples (Figure 4H), as previously observed in histological staining. Regarding the adhesive glycoproteins, such as fibronectin and laminin, there was no statistical difference between the native and decellularized groups (Figure 4I,J). These data confirmed the histological analyses and demonstrated that the produced scaffolds maintained important structural proteins for tissue microarchitecture preservation, as well as adhesive glycoproteins important for cell–ECM interaction. The means of the relative occupied area are summarized in Appendix A.

To evaluate the three-dimensionality of the generated scaffolds, an ultrastructural analysis by scanning electronic microscopy was performed for the native and decellularized tubal segments (Figure 5). It is noteworthy that the decellularization maintained the tubal wall stratification of the dINF–derived scaffolds, with the organized fibers and absence of cells (Figure 5B). In the native infundibulum, there is a dense ciliated secretory epithelium with several epithelial invaginations (Figure 5A,C), while in the decellularized biomaterial, the mucosa portion is completely acellular. At the highest magnification, it was observed that the dINF collagen bundles are organized in a stratified and interspersed manner, with no signs of degradation (Figure 5D). In addition, in the native tissue, there is an intense presence of fimbriae and mucus accumulations (Figure 5E–G), which are absent after the decellularization. In the dINF, the maintenance of thick and thin collagen fiber refinement was also observed, which remained complexly arranged, demonstrating that delicate structures were preserved after the decellularization process (Figure 5F–H).

As expected, the dAMP–derived scaffolds preserved the three tubal wall layers’ maintenance along with the absence of cells (Figure 5J). In the native ampulla, there is the presence of epithelial invagination and mucus (Figure 5I,K,M), which is absent in the acellular scaffolds. At higher magnifications, dense and structured collagen bundles can be observed, demonstrating that the tissue microstructure remained intact in the dAMP (Figure 5N). In addition, the ECM layers were superimposed in a stratified pattern and remained well structured and more distinct in the absence of cells. Similarly to the infundibulum, the fimbria presented in the ampulla epithelium and mucus deposits were abundant in the native tissue (Figure 5O), and absent in the dAMP. Thick and thin collagen fibers were also preserved in acellular scaffolds, with a greater proportion of thick fibers, corroborating the histological findings (Figure 5P). 

The dIST–derived scaffolds presented a more compact and fibrous structure, with the decellularized mucosa and muscular layers more prominent than in the other segments (Figure 5R). In addition, the tissue stratification is well preserved compared to the native tissue. While the native stroma is denser due to the cell anchorage in the ECM components (Figure 5S), in the decellularized material the collagen layers and bundles are more visible, which allowed the observation of a preserved structure (Figure 5T). In the native IST, there was an intense presence of blood vessels surrounded by a dense non-shaped connective tissue (Figure 5Q,U). In the dIST, due to the absence of cells, it is possible to observe round structural scaffolds for the blood vessels (Figure 5V). At higher magnifications, in the IST, the collagen bundles are more interspaced due to cell anchoring (Figure 5W). In the dIST, the collagen bundles are thicker and more agglomerated, which makes the scaffold aspect more fibrous (Figure 5X).

### 3.3. ECM Physico–Chemical Composition Characterization

Spectroscopy emerged as a valuable tool to evaluate biological samples in the bioengineering field, vastly contributing to biomaterial characterization [33]. Vibrational spectroscopy techniques are relatively simple, reproducible, non-damaging, and able to provide valuable information at a molecular level. These techniques also allow the investigation of functional groups, types of chemical bonds, and molecular conformation, leading to direct information about biochemical composition [34,35]. In this study, Fourier Transform Infrared (FTIR-ATR) and Raman spectroscopy were applied to evaluate the physico–chemical alterations in native and decellularized tubal segments.

FTIR spectra (Figure 6A) were previously characterized by [33]. Among the bands, the most noticeable are associated with the amides, which are directly correlated to the collagen spectrum. Amide I can be related to protein secondary structure. Amide II is associated with protein hydration, but also can indicate collagen self-assembly [36,37]. The ratio between the amide III spectrum and the deformation of CH_2_ at 1450 cm^−1^ can be related to the collagen triple helical structure [37]. Finally, the band at 1080 cm^−1^ is associated with the proteoglycan content, which is composed of sulfated and non-sulfated glycosaminoglycans (GAGs) covalently bonded to core proteins [37,38]. Based on the spectra results, to evaluate the level of preservation of ECM molecule content and provide quantitative data after the decellularization process, an integration of band areas of amides I and II, proteoglycans, and the ratio between amide III and 1450 cm^−1^ was performed (Figure 6B–E). There was a statistically significant increase of 22.67% of amide I in the dAMP compared to the native tissue (Figure 6B). On the other hand, in the dAMP and dIST, there was an increase of 26.28% and 24.30%, respectively, in the collagen triple-helical-structure-related band (Figure 6E). No statistically significant alterations for amide II and proteoglycan content was detected after the decellularization compared to the native tissue (Figure 6C,D).

For Raman spectroscopy (Figure 7A), the amide I band has its center at 1665 cm^−1^, while amide III is at 1247 cm^−1^ [38,39,40,41,42,43]. GAG molecules are characteristic at 1063 cm^−1^ and the vibratory mode associated with the phenylalanine aromatic ring is at 1003 cm^−1^ [40,41,43,44,45,46]. The region between 856 and 875 cm^−1^ can be associated directly to proline and hydroxyproline, which are the major amino acids that form the collagen fibers [39,42,43,44,45,46]. Finally, elastin is characterized at 725 cm^−1^ [39,44].

The PCA statistical method was applied to evaluate the spectral differentiation of each region independently. Figure 7B–D present the score graphic for the native and decellularized infundibulum, ampulla, and isthmus, respectively. Each PC1 score graphic has a variation of 53.7%, 63.6%, and 40.2% between the regions’ datasets. This result indicates that there is a sample separation tendency, which means that this differentiation does not indicate a full separation between the spectra and may present similar band contributions between both samples despite the existence of a known spectral difference between the native and decellularized samples.

Similar to FTIR-ATR, the Raman spectra had their areas integrated to evaluate the spectral contribution of each functional group. The quantitative data are exposed in Figure 7E,J. Amide I, elastin, and GAG did not present any statistically significant difference between the native and decellularized samples for the three regions. Regarding the amide III spectrum, only in the ampulla region there was an increase of 19.41% in the decellularized compared to the native group. In the proline/hydroxyproline spectrum, there was an increase of 41.13% in the dAMP and 20.51% in the dIST compared to the native samples. Finally, the Raman spectroscopy detected a decrease in the phenylalanine spectra in all regions. Together, the data demonstrate no evident statistical differences, even presenting a slight divergent tendency compared to the FTIR-ATR, predominantly for amide I, demonstrating a corroboration of the data provided by both techniques. Therefore, spectroscopically, it is possible to assume that the major ECM components of the decellularized components were preserved. These data corroborated the structural and ultrastructural findings regarding the maintenance of scaffolds’ ECM integrity after the removal of cells.

### 3.4. Cytocompatibility Evaluation

The ability to allow cell culture is one of the most important properties for a biomaterial, since the interaction between the scaffold and cells is the main characteristic for proper microenvironment establishment [47]. To evaluate the scaffolds’ cytocompatibility, two assays were performed: the resazurin assay, to gather cell viability, and a scaffold–cell adhesion assay. We also performed a viability assay using immortalized HEK293 epithelial cells derived from human embryonic kidney and canine yolk-sac-derived (YS) cells, which are endothelial precursors [31]. These cells were chosen to demonstrate the non-cytotoxic potential of the generated scaffolds and to evaluate the scaffolds’ capacity to support non-tubal and non-porcine cells. Through SEM analysis, it was possible to observe that after 7 days of culture on the scaffolds both cell types were able to interact with the biological matrices, allowing cell anchorage (Figure 8A–R). In all three scaffolds, YS cells were able to adhere to the ECM fibers, presenting a fibroblastoid shape. There were cell–cell interactions in specific areas, but they did not form a layer on the scaffolds. On the other hand, HEK cells not only adhered to the scaffolds, but also formed layers as observed in native tissue. This demonstrated that the matrices allowed the epithelial cells to organize and anchor to the ECM, according to expectations, since the adhesive glycoproteins of the scaffold remained preserved (Figure 8A–R).

For the cytotoxicity assay, these cells were cultured on scaffolds for 10 days in the presence of resazurin, a purple composite that is reduced by mitochondria, becoming a fluorescent pink composite (resofurin) used to check cell viability [48]. The results showed that in all three scaffolds, the cell viability was above 95% after 1, 3, 7, and 10 days, which verified the scaffolds’ non-cytotoxicity (Figure 9). In terms of performance, the dAMP–derived scaffolds presented the best results, with no statistical difference in any of analyzed periods for both cell types (Figure 9B,E). Concerning dINF–derived scaffolds, HEK cells presented a better performance on the scaffolds compared to YS cells (Figure 9A,D). A similar pattern was observed for dIST–derived scaffolds (Figure 9C,F). In general, the scaffolds allowed satisfactory adherence and survival of epithelial cells from different origins, supporting their broad ability to be used as a platform for in vitro cell culture and tissue reconstruction.

## 4. Discussion

Tubal microarchitecture is responsible for supporting an appropriated microenvironment for the beginning of embryonic development, providing structure and molecular signaling for both gametes and the newly generated embryo to be nourished by the specialized cells of the ciliated secretory epithelium and ECM [2,13]. The structural and compositional configuration of each tubal segment ECM is related to its function in the establishment of maternal–embryonic communication [49,50]. The proportion of ECM components, as well as the stiffness or malleability of each tubal tissue, is directly correlated to its function. The infundibulum has a more flexible structure that allows a more dynamic movement of the fimbriae [18,51]. On the other hand, the isthmus has a more rigid and less flexible structure, being the region that connects the uterine tube to the uterus [18]. The ampulla has an intermediate structural configuration compared to the other two, allowing a greater interaction between the embryo and the tubal tissue [52]. Due to these particularities, the development of decellularization protocols is delicate, the standardization of each step being necessary to preserve the ECM without compromising the efficiency of the cell removal processes.

These matrices may have great biotechnological potential, since the ECM is able to modulate important intracellular processes through the interaction with transmembrane receptors as integrins, as well as influence cytoskeleton reorganization during cell migration [53]. Since they may provide molecular substrate for in vitro models production and supplementary condition for embryo culture, the establishment of proper scaffolds and matrices is essential for biotechnology of reproduction [14,15,33]. Despite these indicatives, only a few studies previously used decellularized tubal matrices for experimental assays until recently [28,54]. Francés-Herrero et al. [28] used decellularized uterine tubes from hormonally stimulated rabbits to develop a hydrogel that could be applied as a supplement for an in vitro embryo culture. Despite their pioneering and success in stablishing acellular tubal matrices, the study did not considered the anatomic distinction of tubal segments, decellularizing the tube as whole, and consequently facing some challenges [28]. This methodology presented some loss of major tubal ECM proteins, which is demonstrated by the GAG general content degradation and the decrease in the identification of some proteoglycan and glycoproteins by proteomic analysis. In comparison, our data revealed that there was a preservation of the main ECM components for the three tubal segments once they were decellularized separately according to tissue density. The degradation of the rabbit tubal ECM reflected directly on the hydrogel bioactivity, since the loss of some bioactive peptides originally present in tubal microenvironment could compromise the mimicking of in vivo conditions [55,56]. Despite the limitations, the use of acellular tube-derived hydrogels was able to induce changes in the metabolic profile of the embryos, which revealed that the ECM components interacted with the embryo in its early stages, allowing an adequate in vitro development with a satisfactory metabolic performance [28].

In a different study conducted by Venkata et al. [54], tubal and uterine organoids derived from human fetal and adult cells were produced and transplanted in tubal and uterine three-dimensional scaffolds to develop an in vitro model that allows the study of the molecular mechanisms related with Muller duct development anomalies. The results demonstrated differences in the proteomic composition of the organoids, and the fetal ones were able to develop properly with Wnt ligand supplementation in the medium [54]. This study innovates on the development of tubal organoids harvested on biological matrices, providing more complexity to the system; however, the protocol used to obtain the tubal acellular matrices differs from our results. Once the tissues were exposed to a high concentration of SDS detergent for ten days, the ECM molecular composition was very likely affected and [57,58,59] due to the lack of an acute analysis of the scaffolds’ structural and compositional characterization, it is inviable to compare this model to the one we presented in this study.

Besides the methodological parameters for the generation of preserved acellular scaffolds, the biological material source is an important variable to be considered in the development of technologies for both experimental assays and large scale applications [60,61]. Pre-pubertal porcine uterine tubes were chosen in this study since pig slaughter occurs with high frequency in commercial slaughterhouses and reproductive tissues are discarded or underutilized, which reinforce the relevance of recycling these biological materials for biotechnological purposes [62,63]. Another interesting factor to be considered is the homogeneity of the samples used to generate the biomaterials, due to the animal’s reproductive cycle consistency [64,65]. Indeed, hormonal stimulation is able to induce structural and functional alteration of reproductive tissues, which reflects on ECM organization and composition, and thus, on the production of biomaterial [66,67,68]. Consequently, the translational potential of porcine-tissue-derived biomaterials is considerable for in vitro reproduction technologies for both humans and other species of economic interest [33].

The maintenance of the matrix scaffolds’ three-dimensional structure is another key factor for the development of more trustworthy in vitro models that mimic a suitable microenvironment for pre-implanted embryos [69,70]. It is noteworthy that 3D culture systems provide adequate conditions for cell culture compared to the traditional 2D ones, allowing greater cell–cell and cell–environment interaction, which impacts on the cellular performance, gene expression, and protein synthesis [71]. Besides the three-dimensionality of the tubal microenvironment, the biomechanical forces that the conceptus is exposed to are important for the embryo culture, acting directly on cell viability and embryo survival [14,15]. Physiologically, inside the uterine tubes and in direct contact with the tubal epithelium, the trophectoderm cells of the conceptus are exposed to mechanical action such as shear stresses and compression, which are able to modulate important intracellular events [72,73,74], and their excessive application may induce cell gene expression alterations in the embryo [72,75]. This aspect of biomaterial production is an interesting challenge for the field.

To compose a complex system, the 3D structure must be cytocompatible to provide the required conditions for several cell types to repopulate the scaffold and allow a tissue reorganization similar to those found in vivo [76,77]. Although there are several cytocompatible biomaterials, many of them lack adhesive molecules that allow suitable cell anchorage and proper interaction between cells and the environment, which is crucial for signaling and structure maintenance [78,79]. Cell establishment and interaction with scaffolds are also related not only to composition, but tissue stiffness, which may explain the different results for the three tubal-derived scaffolds [33]. Our results demonstrate that the decellularized ampulla-derived matrix (dAMP) presented the best performance for cell culture amongst the obtained biomaterials from the three tubal regions. This result might be related to the dAMP’s intermediate structural pattern, which allowed a more appropriate interaction between cells and the ECM. This finding is interesting for reproducibility due to the availability of material, since the ampulla is the most extensive tubal region, besides being the area with which the early embryo interacts for longer periods [13,80,81]. The matrices’ ability to support non-tubal native cells and those from other species is also very important for biotechnological purposes, presenting significant translational characteristics [33]. Altogether, this data proposed decellularization protocols that are efficient to promote a balance between cell removal and ECM preservation for the three uterine tube regions. The produced matrices also presented great cytocompatibility, allowing both cell populations’ viability in vitro.

## 5. Conclusions

This study was the first to present a structural and compositional characterization of decellularized scaffolds derived from the three porcine tubal regions. The generated scaffolds remained three-dimensionally structured and with the major ECM components preserved. In addition, their cytocompatibility was verified, since they allowed cell survival and provided the conditions for cell adhesion and integration with the scaffolds. These findings demonstrated that matrices might be used as a valuable tool for future applications as supplements for specific embryo culture mediums or even substrates for 3D bioprintable hydrogel production. Such biotechnologies can be used for regenerative therapies in the female reproductive tract or for tubal organoid production to perform drug screening tests or even to study unknown molecular mechanisms related to infertility in the pre-implantational stages in humans and high-interest commercial species. Given the biological and molecular role of the extracellular matrix in the distinct uterine tube portions, these decellularized region-specific matrices may be suitable bioactive compounds for a great number of applications in cell and embryo culture, which provide unique molecular stimulation, enhancing the system complexity parameters to those found in vivo.

## Figures and Tables

**Figure 1 biomimetics-09-00382-f001:**
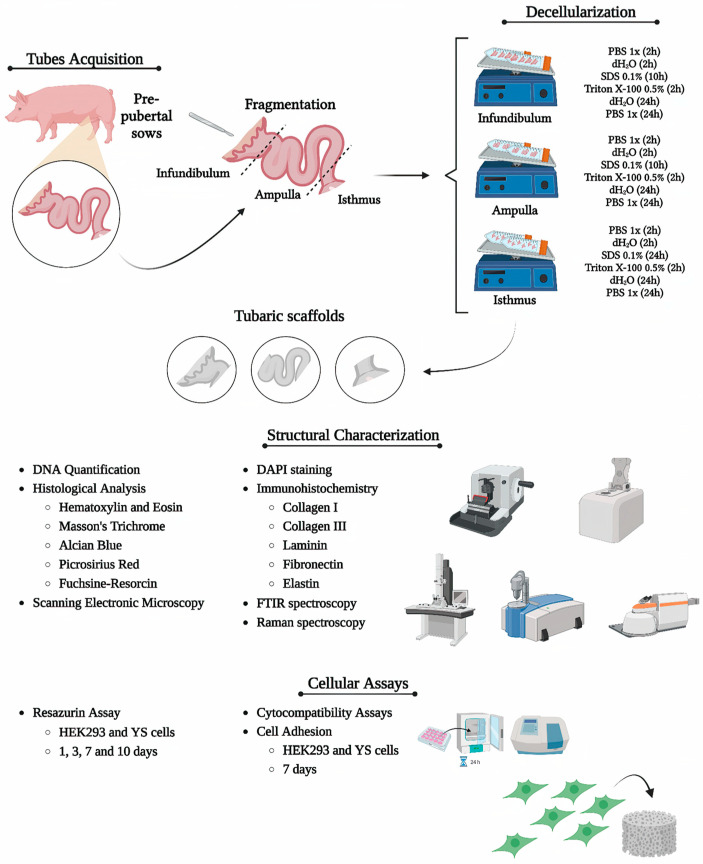
Representative scheme demonstrating the methodology used in the study. Created with BioRender.com (18 June 2024).

**Figure 2 biomimetics-09-00382-f002:**
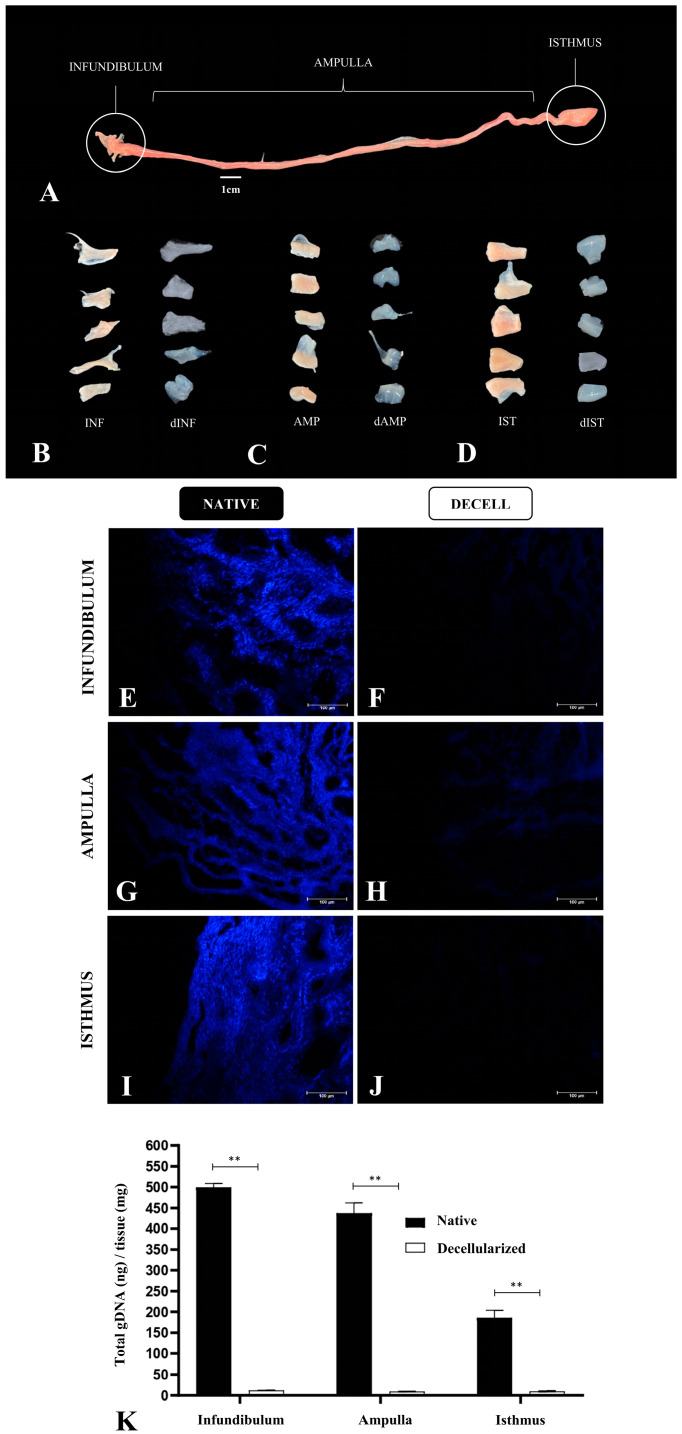
Decellularization of porcine fallopian tube segments. An anatomic division of the porcine fallopian tube, highlighting the three segments: infundibulum, ampulla, and isthmus (**A**). Comparison of native infundibulum (INF) and decellularized infundibulum scaffolds (dINF) (**B**). Comparison of native ampulla (AMP) and decellularized ampulla scaffolds (dAMP) (**C**). Comparison of native isthmus (IST) and decellularized isthmus scaffolds (dIST) (**D**). DAPI staining of native tubal segments (**E**,**G**,**I**) and decellularized scaffolds (**F**,**H**,**J**). DNA quantification of tubal tissues before and after the decellularization process (**K**). ** *p* < 0.05 compared to the native tissue.

**Figure 3 biomimetics-09-00382-f003:**
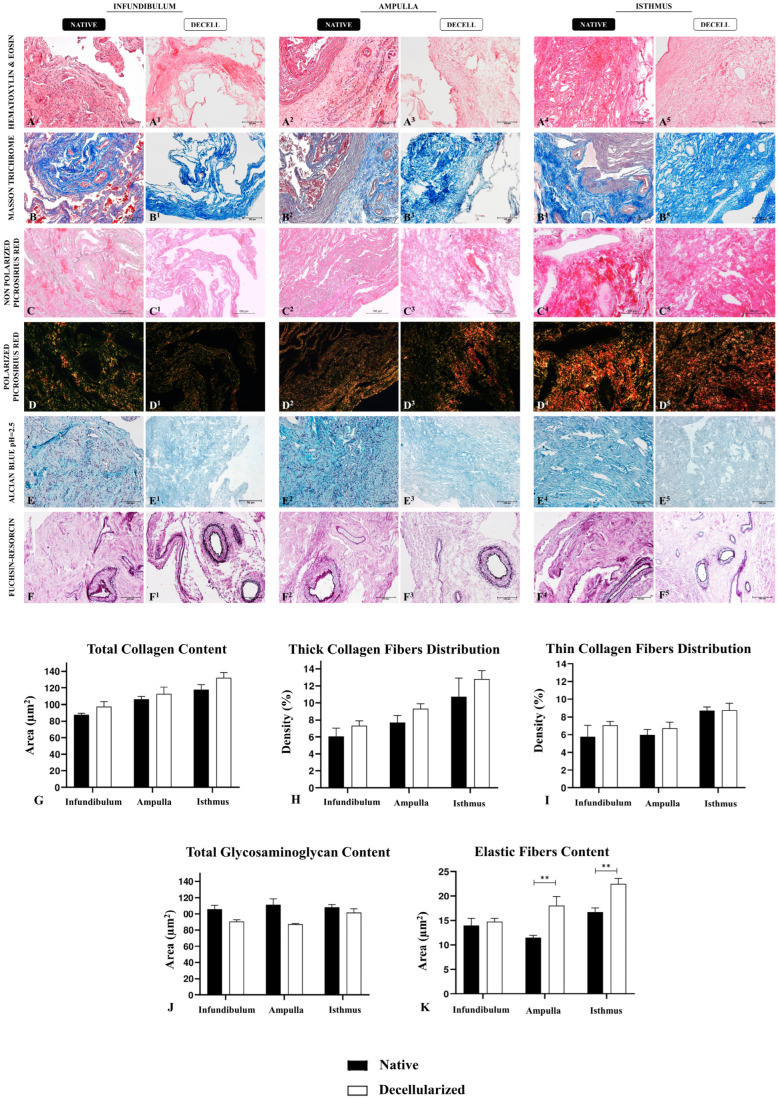
The histological analysis of the tubal ECM from the native and decellularized infundibulum, ampulla, and isthmus. Hematoxylin and eosin staining indicates the tissues’ general structure and presence or absence of cells in native and decellularized samples (**A**–**A^5^**). Masson’s trichrome staining highlights the total collagen content in blue in native and decellularized samples (**B**–**B^5^**). Picrosirius red staining also indicates collagen content (**C**–**C^5^**). Polarized Picrosirius red staining distinguishes mature collagen (red and yellow) and immature collagen (green) (**D**–**D^5^**). Alcian blue staining evidences GAG content in native and decellularized samples (**E**–**E^5^**). Weigert’s fuchsin–resorcin staining highlights elastic fibers content in native and decellularized samples (**F**–**F^5^**). Quantitative analysis of the relative occupied area of the general ECM components in native and decellularized porcine infundibulum, ampulla, and isthmus. Comparison of total collagen content (**G**), thick collagen fiber distribution (**H**), thin collagen fiber distribution (**I**), total glycosaminoglycan content (**J**) and elastic fiber content (**K**) between the native and decellularized tubal samples. ** Statistically significant with *p* < 0.05 when compared with native group.

**Figure 4 biomimetics-09-00382-f004:**
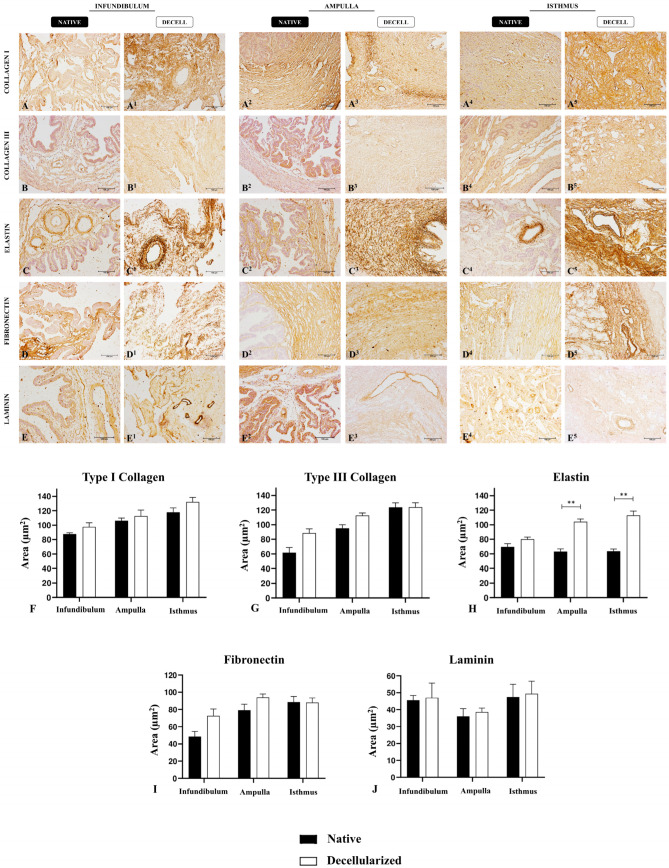
Immunohistochemical analysis of the main ECM components of the native and decellularized infundibulum, ampulla, and isthmus. Immunolocalization of type I collagen (**A**–**A^5^**), type III collagen **(B**–**B^5^)**, elastin (**C**–**C^5^**), fibronectin (**D**–**D^5^**), and laminin (**E**–**E^5^**) in native and decellularized samples. Quantitative analysis of the relative occupied area of the main ECM components in native and decellularized porcine infundibulum, ampulla, and isthmus. Comparison of type I collagen content (**F**), type III collagen content (**G**), elastin content (**H**), fibronectin content (**I**), and laminin content (**J**) between the native and decellularized tubal samples. ** Statistically significant with *p* < 0.05 when compared with native group.

**Figure 5 biomimetics-09-00382-f005:**
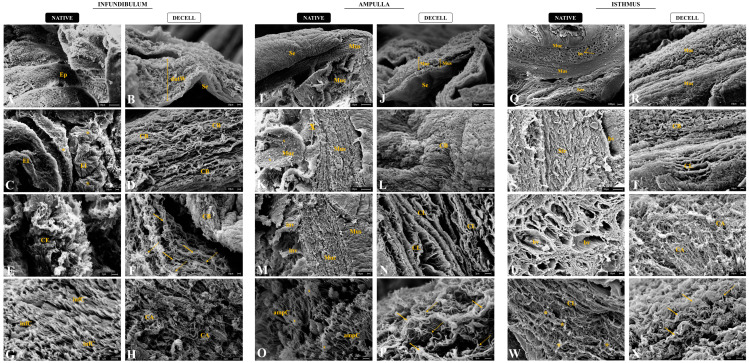
Scanning electronic micrographs of the native and decellularized infundibulum (**A**–**H**), ampulla (**I**–**P**), and isthmus (**Q**–**X**). Ep (epithelium), dutW (decellularized uterine tubal wall), Muc (mucosa), Mus (*muscularis*), Se (serous), EI (epithelial invaginations), CE (ciliated epithelium), CB (collagen bundles), infC (cilia of the infundibulum), CA (collagen arrangements), inv (invagination), CL (collagen layer), ampC (cilia of the ampulla), bv (blood vessels), Str (stroma). Full yellow arrows indicate thick fibers. Dotted arrows indicate thin fibers. Yellow asterisks indicate cells among ECM fibers. Orange “x” indicates presence of mucus.

**Figure 6 biomimetics-09-00382-f006:**
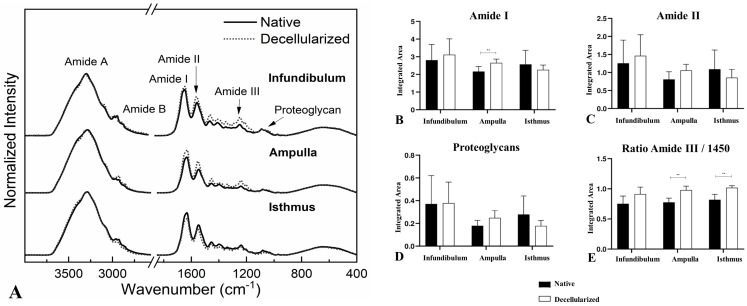
FTIR-ATR spectra from the regions of the native and decellularized infundibulum, ampulla, and isthmus (**A**). Integrated band areas of amide I (**B**), amide II (**C**), proteoglycan content (**D**), and band area ratio of amide III: 1450 cm^−1^ (triple helical structure of collagens) (**E**) in the native and decellularized porcine infundibulum, ampulla, and isthmus. ** Statistically significant with *p* < 0.05 when compared with native group.

**Figure 7 biomimetics-09-00382-f007:**
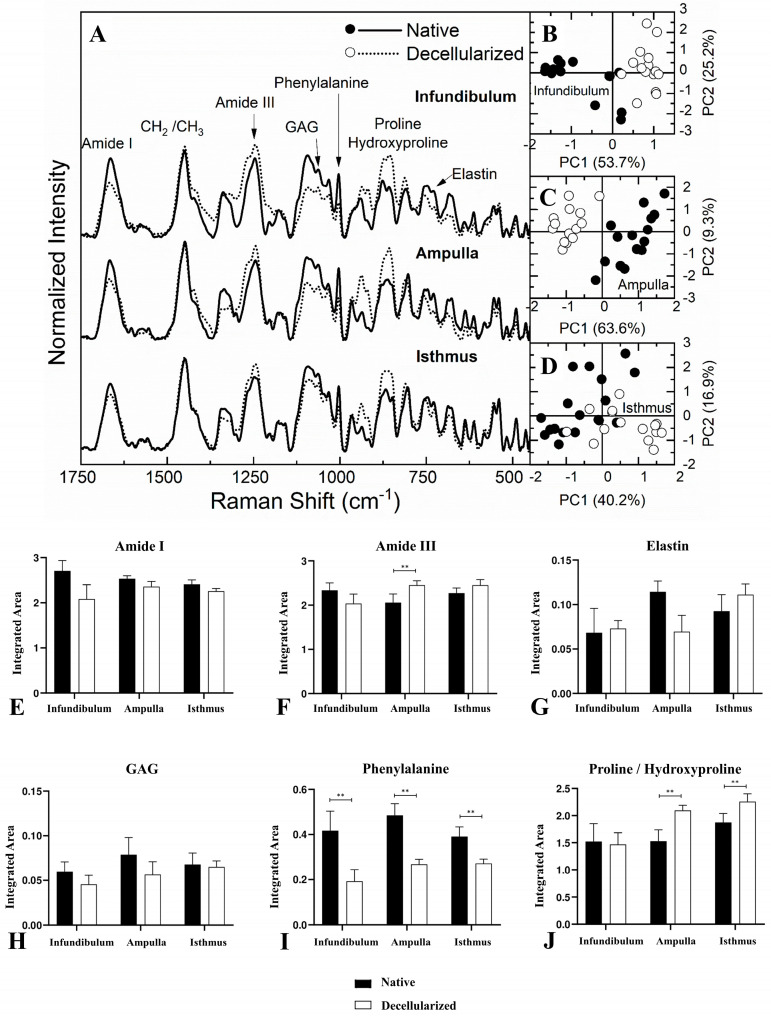
Average Raman spectroscopy spectrum of the regions for the native and decellularized samples from the porcine fallopian tube (**A**). Principal component analysis (PCA) score graph obtained for the infundibulum region (**B**). PCA core graph for the ampulla region (**C**). PCA core graph for the isthmus region (**D**). Integrated band areas of amide I (**E**), amide III (**F**), elastin (**G**), GAG content (**H**), phenylalanine (**I**), and proline/hydroxyproline (**J**) in the native and decellularized porcine infundibulum, ampulla, and isthmus. ** Statistically significant with *p* < 0.05 when compared with native group.

**Figure 8 biomimetics-09-00382-f008:**
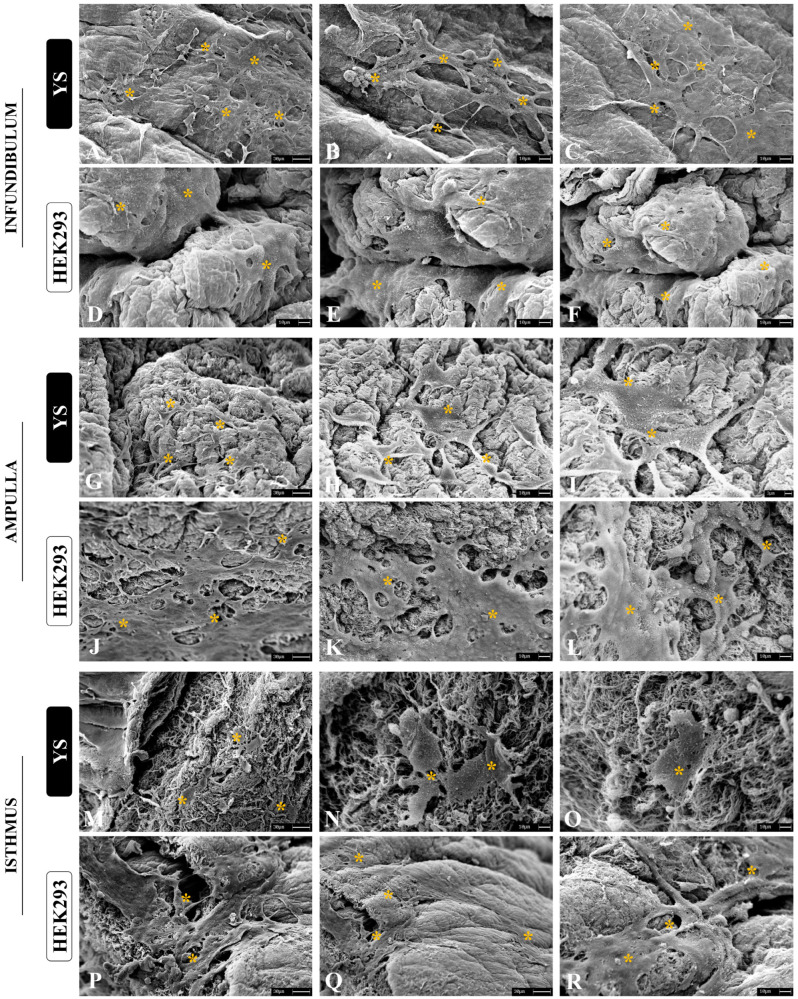
Scanning electronic micrographs of the adhesion assay performed with YS and HEK293 cells seeded on decellularized infundibulum, ampulla, and isthmus scaffolds after 7 days (**A**–**R**). Orange asterisks indicate the cells anchored on the scaffolds’ ECM fibers.

**Figure 9 biomimetics-09-00382-f009:**
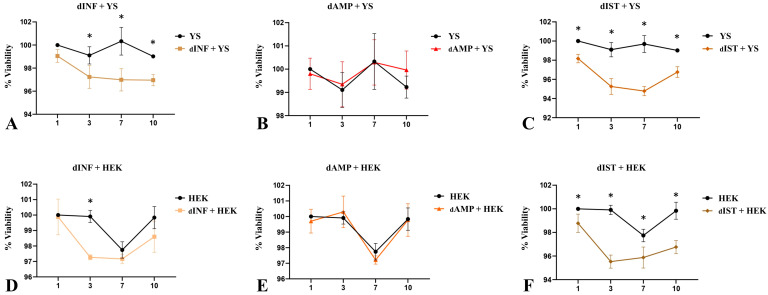
Cell viability assessment of YS and HEK293 cells cultured on decellularized infundibulum (dINF), ampulla (dAMP), and isthmus (dIST) after 1, 3, 7, and 10 days. Trend graphs of YS cells (**A**–**C**) and HEK293 cells (**D**–**F**). The obtained absorbance data were converted and expressed as a viability percentage. * Statistically significant for *p* < 0.01 when compared to the control group.

## Data Availability

All data related to this study are available in the article. Additional raw datasets generated during the study are too large to be shared publicly but are available upon request from the corresponding author.

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
