# Peer review of "Region-Specific Decellularization of Porcine Uterine Tube Extracellular Matrix: A New Approach for Reproductive Tissue-Engineering Applications"

_biomimetics, 2024, doi:10.3390/biomimetics9070382_

Round 1
Reviewer 1 Report
Comments and Suggestions for Authors
In the manuscript "Region-Specific Decellularization of Porcine Uterine Tube Extracellular Matrix: A New Approach for Reproductive Tissue Engineering Applications" the authors present protocols to obtain decellularized scaffolds derived from porcine infundibulum, ampulla, and isthmus to provide suitable sources of biomaterials for tissue-engineering approaches. This study is interesting and is the first to describe the structural and compositional characterization of scaffolds obtained from three porcine tubal regions. The results are very thoroughly processed and presented.
I have minor comments/questions:
- - Please provide information of PBS molarity and pH,
- - Line 133 please correct sodium dodecyl phosphate in sodium dodecyl sulphate,
- - Is there a washing step between SDS and Triton X-100?,
- - scale bars on figures are barely visible (I don't know if it's up to my computer), please specify their lengths in the figure legends (Fig. 3 and 4),
- - In Figures 5 and 8, please highlight the yellow markers (asterisks and text) more prominently, if possible,
- - Lines 393 and 396 please correct e to and,
- - From lines 604 to 608, several technical errors are observed (double periods, missing spaces, double spaces, etc.), please correct them,
- - Line 628, two commas,
- - Line 669 correct in vivo to italic.
Author Response
June 12th, 2024.
Editor-in-Chief
Biomimetics
Dear editor,
Please find enclosed the revised version of our manuscript entitled “Region-Specific Decellularization of Porcine Uterine Tube Extracellular Matrix: A New Approach for Reproductive Tissue-Engineering Applications” submitted to Biomimetics.
We addressed all of the reviewer’s comments, which were extremely relevant for improving our manuscript. A point-by-point answer is below addressed:
Reviewer #1: “Please provide information of PBS molarity and pH,”
Answer: Thank you for the suggestion. We added this information in the text (line 133).
Reviewer #1: “Line 133 please correct sodium dodecyl phosphate in sodium dodecyl sulphate,”
Answer: Thank you for the suggestion. We altered in the manuscript.
Reviewer #1: “Is there a washing step between SDS and Triton X-100?”
Answer: No, we washed the scaffolds with Triton X-100 immediately after the SDS, once the interaction between an anionic and non-anionic detergent would improve its removal.
Reviewer #1: “scale bars on figures are barely visible (I don't know if it's up to my computer), please specify their lengths in the figure legends (Fig. 3 and 4),”
Answer: Thank you for the suggestion. The uploaded PDF file decreases the quality of the figures, which may have impaired your visualization, but I assure that, in the original figures, the scale bars are visible.
Reviewer #1: “In Figures 5 and 8, please highlight the yellow markers (asterisks and text) more prominently, if possible,”
Answer: Thank you for the suggestion.
Reviewer #1: “Lines 393 and 396 please correct e to and,”
Answer: Thank you for the suggestion. We altered in the manuscript.
Reviewer #1: “From lines 604 to 608, several technical errors are observed (double periods, missing spaces, double spaces, etc.), please correct them,”
Answer: Thank you for the suggestion. We altered in the manuscript.
Reviewer #1: “Line 628, two commas,”
Answer: Thank you for the suggestion. We altered in the manuscript.
Reviewer #1: “Line 669 correct in vivo to italic.”
Answer: Thank you for the suggestion. We altered in the manuscript.
I hope you find the revised version of our manuscript suitable for publication. Thank you in advance for your consideration.
Sincerely,
Prof. Dr. Ana Claudia Oliveira Carreira
Faculty of Veterinary Medicine and Animal Science
University of São Paulo
Reviewer 2 Report
Comments and Suggestions for Authors
The group established region-specific protocol to obtain decellularized scaffolds from the porcine uterine tubes. Using various assays, the group validated the decellularization process showing there was reduction of cellular content and characterized the protein contents. Overall the study was well-presented and have some minor suggestions.
- For the different regions of the uterine tube, consider colour coding the bar graphs.
- In figure 9A, C, F since there were significant differences in the viability of the cells after cultured with the scaffold. A more detailed discussion is required to address these observations. Refrain from stating the cell survived with the scaffold in the abstract.
Comments on the Quality of English LanguageMinor mistakes and formatting need revising.
Author Response
June 12th, 2024.
Editor-in-Chief
Biomimetics
Dear editor,
Please find enclosed the revised version of our manuscript entitled “Region-Specific Decellularization of Porcine Uterine Tube Extracellular Matrix: A New Approach for Reproductive Tissue-Engineering Applications” submitted to Biomimetics.
We addressed all of the reviewer’s comments, which were extremely relevant for improving our manuscript. A point-by-point answer is below addressed:
Reviewer #2: “For the different regions of the uterine tube, consider colour coding the bar graphs.”
Answer: Thank you for the suggestion. We believe that the current colors represent the native and decellularized tissues, which follows an aesthetical pattern of our lab articles, so we chose to leave black and white.
Reviewer #2: “In figure 9A, C, F since there were significant differences in the viability of the cells after cultured with the scaffold. A more detailed discussion is required to address these observations. Refrain from stating the cell survived with the scaffold in the abstract.”
Answer: Thank you for the suggestion. We altered in the abstract this information, replacing for “allowing high cell viability rates”. We improved the discussion related to these differences in cell viability rates.
I hope you find the revised version of our manuscript suitable for publication. Thank you in advance for your consideration.
Sincerely,
Prof. Dr. Ana Claudia Oliveira Carreira
Faculty of Veterinary Medicine and Animal Science
University of São Paulo